# Energy and structural evolution process of high-altitude and long-runout landslides induced by a strong earthquake

Yunfeng Ge1\*,2, Bin Hu1, Huiming Tang1,3, Xiaodong Fu4, Lei Zhu5

- <sup>1</sup>Faculty of Engineering, China University of Geosciences, Wuhan, 430074, China <sup>2</sup>School of Natural Resources Science and Technology, Xinjiang University of Technology, Hotan, Xinjiang 848000, China
  - <sup>3</sup>Three Gorges Research Center for Geo-hazard, China University of Geosciences, Wuhan, 430074, China
- 10 <sup>4</sup>State Key Laboratory of Geomechanics and Geotechnical Engineering, Institute of Rock and Soil Mechanics, Chinese Academy of Sciences, Wuhan 430071, China
  - <sup>5</sup>Key Laboratory of Mountain Hazards and Surface Process, Institute of Mountain Hazards and Environment, Chinese Academy of Sciences, Chengdu, 610041, China
  - Correspondence to: Yunfeng Ge (geyunfeng@cug.edu.cn)
- 15 Abstract. It is often difficult to restore the evolution process and energy transfer of ancient landslides that have occurred over a long time. In this paper, the contour restoration method is used to restore the topography before the Mogangling landslide according to the contour of the surrounding mountains. In order to better analyze the landslide movement process and analyze the energy change, the numerical simulation method is used to reproduce the Mogangling landslide. In the process of numerical simulation, the displacement and velocity of the whole and part of the landslide mass are monitored, respectively, so as to extract their potential energy, kinetic energy, and dissipation energy. At the same time, the effective collision of blocks in the process of landslide is obtained by extracting the local peak value of the displacement velocity curve. Meanwhile, the Alpha shape algorithm was employed to extract the volume and surface area of the landslide body from 3D point cloud data, thereby enabling 25 the calculation of its volume expansion rate and area growth rate to quantify the morphological evolution characteristics during the landslide movement process. The results show that for the whole landslide, its energy change conforms to the law of conservation of energy; For some blocks, the energy is not conserved due to the collision and compression of surrounding rock mass; Compared with the upper rock mass, the lower rock mass is compressed more frequently, receives more energy transfer, 30 and has a longer migration distance.

## 1 Introduction

35

Landslide is a common and widespread geological disaster, which causes great loss of life and property. In recent years, the chain disasters of high-level rock landslides in high-intensity earthquake areas have shown a trend of frequent occurrence (Carpignano et al., 2009; Fan et al., 2019a), such as donghekou landslide debris flow dammed lake induced by the 2008 Wenchuan earthquake (Sun et al., 2011; Wang et al., 2014; Wang, 2021), Hongshiyan landslide induced by the 2014 Ludian earthquake (Hu et al., 2017; Tian et al., 2017), which has seriously affected people's lives and property and the safety of major projects. In this study, the Mogangling landslide that occurred in 1786 was repaired and

55

reconstructed to study the formation mechanism and energy transfer process of high-level rock 40 landslide induced by earthquake, so as to summarize the law of high-level rock landslide induced by earthquake and reduce the impact of landslide on life and property.

At present, in terms of surface terrain restoration, the terrain restoration before a landslide is less. For landslides that have not occurred too far away, the key information of landslide characteristics is often obtained through photos, field surveys, optical remote sensing images and UAV impact, so as to further build geomorphic maps through GIS (Geographic Information System) for subsequent analysis (Zhang et al., 2024; Song at al.2017; Fan et al., 2020; Wu et al., 2024). For most numerical simulations involving terrain restoration, a two-dimensional simulation is more often used. The simplified model is adopted to simulate the overall process of landslide by inputting external stress (Du et al., 2021; Cheng 2014; Li et al., 2024). This paper uses the method of restoring the continuity of contour lines to establish the terrain data before a landslide, which can provide a new idea of terrain restoration before a landslide for the same type of research.

Several research studies have been conducted to clarify the genesis and propagation mode of this kind of landslide event, and some reasonable methods have been proposed (Huang et al., 2012; Intrieri et al., 2018). For field investigations, based on satellite remote sensing, geophysical drilling, electrical resistivity tomography (ERT), and drone aerial photography, the triggering mechanism, runout behavior, and sedimentation characteristics of the landslide can be detected (Huang et al., 2019; Fan et al., 2019; Zhao et al., 2020). Meanwhile, multiple numerical methods and drone video have been employed to co-analyze the dynamic characteristics of landslides well (Zhu et al., 2010; Zhang et al., 2020a, b; Cuomo et al., 2021; Shugar et al., 2021). These efforts have been made to better understand the dynamic process of rockslide-debris avalanches. Due to the suddenness of long-runout landslides, direct observation data are rarely available in previous studies. In recent years, landslide seismology has taken on more visibility (Navarre et al., 2009; Van Herwijnen and Schweizer, 2011; Helmstetter and Garambois, 2010; Cook et al., 2021; Pandey et al., 2021). However, they still lack a quantitative and systematic study of this topic, especially on energy evolution during the collision.

This study aims to comprehensively utilize field investigations, drone aerial surveying, discrete element numerical simulation, and block energy calculation methods to systematically reconstruct the dynamic evolution process of the Mogangling landslide. On this basis, it focuses on revealing the mechanisms of energy accumulation, transfer, and dissipation during the landslide event. Through quantitative analysis of energy evolution pathways and structural change characteristics, the research elucidates the movement trajectory, spatiotemporal evolution of velocity, and debris flow propagation characteristics of the landslide mass under strong seismic conditions. The findings are intended to provide a theoretical foundation and scientific support for early warning and risk mitigation of high-altitude and long-runout landslides triggered by intense earthquakes.

## 2 Methodologies

In order to describe the energy transfer process of earthquake-induced landslide, the representative Mogangling landslide is selected as a case study. As shown in Fig1., the research method involves




three parts: (1) surface terrain restoration: the terrain before the landslide is deduced, and the landslide model is established based on the existing terrain of the Mogangling landslide by restoring the continuity of contour lines. It includes strata, geological structure, boundary conditions, seismic force input and mechanical parameters; (2) Analysis of Landslide Evolution Process: carry out discrete element numerical simulation of the landslide model, infer the landslide evolution process according to the changes of motion parameters and stress in the numerical simulation process, and add effective collision extraction to analyze the motion law; (3) Energy evolution analysis: the potential energy, kinetic energy and dissipation energy of the landslide are extracted by monitoring the speed and displacement of the whole landslide and some individuals, and their laws are observed. At the same time, through the comparison of effective collision and motion trajectory, the influence of effective collision frequency of different parts of landslide mass on energy transfer is analyzed; (4) Extraction of volume and area of landslide accumulation area: the envelope of landslide mass is generated by alpha shape, so the volume expansion rate and surface area growth rate of landslide mass in the process of landslide can be calculated, and the evolution law can be obtained by combining the energy change.

Fig1. Flow chart of the proposed method of this study






#### 2.1 Geo-mechanical model building

As shown in Fig2., the Mogangling landslide is located at the upstream of Caihong bridge, Jinguang village, Detuo Township, Luding County, Sichuan Province, and the right bank of Dadu River is close to the mouth of Moxi River, a tributary. The coordinates of the landslide center are: 29 ° 37 ′ 30.726 " north latitude, 102 ° 09 ' 41.190 " east longitude. It connects Luding County to the north, Shimian County to the south, Moxi town, and Kangding County to the West. The middle reaches of the Dadu River are located at the intersection of "Y" - shaped faults composed of the Xianshuihe fault, Longmenshan fault, and Anninghe fault. It is an area of high seismic intensity, and nearly 20 large and super large earthquake landslides are densely distributed here (from the Detuo Guza river section). The lithology of the landslide area is mainly gray white, strongly to weakly weathered Jinning plagioclase granite (702). Quartz diorite (O2) and a small amount of basic magmatic rocks are distributed on the western slope of Mogangling. The sand slate of the Triassic Baiguowan formation (T<sub>3bg</sub>) is exposed on the left bank of the Dadu River. The quaternary system mainly includes landslide accumulation (Q<sub>4</sub><sup>del</sup>), colluvial Deluvial deposit (Q4col+dl) and alluvial deposit (Q4al), of which colluvial Deluvial deposit is located at the rear edge of the landslide, the mountain side of the left bank of Dadu River and the surrounding steep slope, and alluvial deposit is located at the river valley terrace. Historical earthquakes have caused extremely heavy losses to southwest China, especially the series of losses caused by earthquakes, landslides, river blocking, and a breach super flood disaster chain. The Mogangling landslide induced by the Kangding earthquake (Ms7 3/4) in 1786 burst the Dadu River after 9 days. The sudden flood made the downstream area a vast ocean, and tens of thousands of people drowned.

In the construction of major projects in Western China, the effect of seismic force on slope instability has been underestimated to a certain extent. For example, the stability investigation of the reservoir slope of large-scale power stations in Western China mostly focuses on the stability of existing landslides and the evaluation of the surging waves after instability, and the analysis of the instability probability of potentially unstable high and steep slopes under the action of seismic force and the evaluation and analysis of dam break or dam overturning caused by abnormal bedrock landslide such as surging waves are relatively few. The Dadu River high-intensity area is a major hydropower project in China. There are a large number of paleoseismic landslides in this area. When the development background and laws of these major earthquake landslides have not been fully studied in detail, the construction of high-risk reservoirs and dams has great potential risks. Therefore, taking the Mogangling landslide as an analysis case, exploring the dynamic propagation characteristics can provide an evaluation basis for the potential landslide in the high-intensity area of the Dadu River, which is of great significance to the development of Western China.



Fig2. Position of the study area: (a) the location of the Mogangling landslide; (b) remote sensing image of Mogangling; and (c) engineering geological map.

In order to study the energy transfer in the process of landslide, numerical simulation technology is used in this study. Before model construction, it is necessary to establish a geometric model of the ground surface to obtain the sliding bed and sliding mass. According to the DEM data obtained by UAV scanning, the point coordinates are extracted, and the landslide area is delimited for precision screening to reduce the point density. Convert the geographical coordinate system to the projection coordinate system to facilitate the export of the filtered point cloud. After reducing the noise caused by vegetation and repairing the cavity, the complete topographic surface data of the Mogangling area after the landslide were obtained. The obtained terrain data is corrected by the normal vector to generate the existing model as the sliding bed. Because the landslide occurred too long ago, only the Tiezhuang Temple tablet unearthed in the 1980s has records of earthquake landslide, river dam break, and water discharge. Therefore, this study uses the method of restoring contour lines to simulate the terrain before the landslide. The scanned terrain data is output as contour lines, and then the contour lines corresponding to the sliding bed and accumulation area are modified to be straight. The restored





contour lines are transformed into point clouds for subsequent processing, and the topographic map before the landslide is obtained after noise reduction, cavity repair, normal correction, and other processing.

As shown in Fig3.(a), the geometric model of the terrain surface is established according to DEM. The whole landslide occurrence area can be divided into four parts: the rock wall area, colluvial slope area, slope debris flow area, and main accumulation area. There is a reverse fault along the right bank of the Dadu River, namely the Dadu River fault (Detuo fault). According to the reconstructed topographical surface geometry, the Mogangling landslide is a typical armchair-shaped terrain, with a main sliding direction of 75°, distributed along the NEE-SWW direction, about 450 m long, about 1000 m wide along the river, and a planar area of 0.45 km<sup>2</sup>. The average thickness of the landslide mass is about 100 m, and the volume is about  $2400 \times 104$  m<sup>3</sup>. The elevation of the slope toe is 1120 m (about 5 m higher than the river surface), and 1120 m-1330m is the front edge of the landslide accumulation mass, with a slope of about 50°; The elevation of landslide platform is between 1330 m and 1380 m, and the slope is about 5-12°; The elevation of the rear wall of the landslide is between 1380 m and 1890 m, of which 1380 m to 1600 m are covered by colluvial deposits, with a slope of about 35°, and 1600 m to 1890 m are landslide walls, with a slope of about 57°; The elevation of the back wall ridge is about 2000 m. The contour line is modified along the extension direction of the mountains on the left and right sides to repair the sliding bed generated by the landslide into the terrain of the mountains. The modified contour is converted into a point cloud and repaired. As shown in Fig3. (b), make a section along a-a'and b-b' in Fig2. (c). The results are shown in Fig3. (c). It can be seen that although compared with the real terrain, the restored terrain is smooth and the surface does not fluctuate, the mountains become continuous, which can be regarded as the terrain before the landslide.




Fig3. Landslide restoration diagram: (a) 3D views of the study area; (b) Engineering geological profile; (c) Comparison map of surface topography before and after restoration

Due to the need to simulate the large deformation and displacement of rock mass in the process of landslide, the three-dimensional discrete element program (3DEC), one of the most widely used numerical methods in rock mechanics, is used to simulate the motion process of the Mogangling landslide. According to the extracted terrain data after the landslide, the three-dimensional modeling is carried out, and the landslide mass is constructed according to the restored terrain surface.

As shown in Fig4., the established model includes the landslide mass and base block. The length of the landslide mass in the sliding direction is 1021 m, and the vertical height is about 747 m; the bottom of the landslide mass is 179 m away from the Dadu River, and the vertical height is about 131 m. Because the occurrence distribution data of the structural plane of the landslide mass before the landslide can not be obtained, the impact of the structural plane on the landslide process has to be ignored in this study. The landslide mass is evenly divided into 8.7 m×8.7 m blocks on the xoy surface. The height is determined by the vertical height from the surface of the landslide mass to the sliding bed. A uniform structural plane is set inside. According to the geological engineering drawing, different strata are generated and marked with different colors. The whole model contains 25345 blocks, 2788 m long, 3341 m wide, 1121m high, and the volume of the landslide is 317563 m³.




Note that, since the Mogangling landslide was triggered by the 1786 Kangding earthquake, the considerable time elapsed since the event and the lack of instrumental ground-motion records for this historical earthquake make it impossible to directly obtain the seismic input parameters from the original incident. It is worth noting that both the Mogangling landslide and the Wenchuan earthquake zone belong to the tectonic system of the Longmenshan Fault Zone. Extensive research indicates that the Kangding earthquake is comparable to the 2008 Wenchuan earthquake in terms of damage distribution, affected intensity, and estimated magnitude. Therefore, this study adopts the acceleration records measured at the Luding station during the Wenchuan earthquake as the input ground motion for the dynamic numerical simulation analysis of the Mogangling landslide.

Fig4. Discrete numerical model with fine topography

In the course of a discrete-element simulation-particularly when modeling nominally semi-infinite bodies such as the use of fixed or elastic boundaries in a static analysis will inevitably reflect outbound seismic waves into the model during a subsequent dynamic analysis. Consequently, when employing 3DEC for seismic-dynamic modeling, it is essential to impose non-reflecting viscous boundaries in conjunction with free-field boundaries. Specifically, viscous boundaries are applied at the model base where the incident seismic waves enter, while free-field boundaries constrain the lateral faces that are not subjected to seismic input. Dynamic loading must be introduced only after the model has attained static equilibrium; i.e., the dynamic analysis is predicated on a prior static analysis that has reached convergence. For the static phase, boundary conditions consist of vertical-direction velocity restraints at the base and horizontal-direction velocity restraints along the lateral faces.

In the numerical simulations, the blocks were assigned a Mohr-Coulomb constitutive model. The physical and mechanical properties of both the rock mass and the discontinuities critically influence the

deformation-failure mechanisms of the rock slope and the evolution of key response variables such as displacement, velocity, and acceleration. In this study, the selection of physical-mechanical parameters was based primarily on laboratory testing, empirical analogy, and parameter inversion. The slope is underlain by plagioclase granite, which is subdivided into strongly weathered and moderately weathered zones. The discontinuities are represented by both fully developed potential rupture planes and joint sets, all of which are governed by the Mohr-Coulomb failure criterion. In the numerical model, the parameters for the rock mass and discontinuities used in the analyses are those listed in Table 1.

Table 1 Physical parameters of rock mass and discontinuities

| Rock mass                 | Density    | Cohesion | Friction | Bulk             | Shear           |  |  |
|---------------------------|------------|----------|----------|------------------|-----------------|--|--|
| ROCK IIIass               | $(kg/m^3)$ | (MPa)    | (°)      | (GPa)            | (GPa)           |  |  |
| bed rock                  | 2700       | 5.35     | 50       | 21.90            | 15.10           |  |  |
| landslide mass            | 2300       | 3.25     | 35       | 11.90            | 6.9             |  |  |
| Discontinuities           | Tension    | Cohesion | Friction | Stiffness-normal | Stiffness-shear |  |  |
|                           | (MPa)      | (MPa)    | (°)      | (GPa/m)          | (GPa/m)         |  |  |
| discontinuities           | 0.03       | 0.8      | 15       | 960              | 1500            |  |  |
| potential discontinuities | 0.01       | 1.1      | 10       | 100              | 180             |  |  |

### 2.2 Effective collision identification

230

During the landslide, numerous block collisions occurred, altering the kinematic behavior of the rock mass. Some impacts produced substantial acceleration of individual blocks, while others caused deceleration due to energy dissipation. To elucidate the anomalous dilation of the sliding mass, this study examines those collisions that markedly increase block velocities-hereafter termed "effective collisions".

Effective collisions of rock blocks are identified by their characteristic velocity changes. Fig5. (a) presents a typical velocity-travel-distance curve for an individual block. A local velocity peak (denoted by red or purple markers) corresponds to an acceleration phase, but only those pronounced peaks (highlighted in red) yield a substantial velocity increase and, consequently, a markedly longer travel distance. Hence, effective collisions can be detected by pinpointing the significant local maxima on the velocity-distance profile.

Significant local maxima are defined as points whose velocity markedly exceeds that of their neighbors. To pinpoint these prominent peaks on the velocity-distance curve, a suitable prominence threshold must be chosen-one that quantifies each peak's standout height, width, and position relative to adjacent peaks. This threshold critically influences the detection of effective collisions. As illustrated in Fig5. (b-f), low thresholds (10% and 20%) spuriously flag minor undulations as peaks, leading to an overestimation of effective collisions, whereas high thresholds (40% and 50%) miss important local maxima and thus undercount them. At a threshold of 30%, however, the identification of effective collisions is deemed acceptable. Under this setting, 1,139 effective collisions were detected from the travel-distance versus velocity profile.

Fig5. Effective collisions identification: (a) definition of effective collisions; (b) threshold=10%; (c) threshold=20%; (d) threshold=30%; (e) threshold=40% and (f) threshold=50%.

# 240 2.3 Energy evolution

During a landslide, dynamic energy propagation involves three forms of energy: kinetic energy (KE), potential energy (PE), and dissipated energy (DE), with the sum of KE and PE defined as mechanical energy (ME). For an individual rock block within the sliding mass (i.e., a single block in the numerical model), at time  $t_{i_i}$  its mechanical energy is given by:

$$245 ME(t_i) = KE(t_i) + PE(t_i) (1)$$

where, at time  $t_i$ ,  $PE(t_i)$ , and  $DE(t_i)$  can be expressed as:

$$KE(t_i) = \frac{1}{2}m[v(t_i)]^2$$
 (2)

$$PE(t_i) = mgh(t_i)$$
(3)

$$DE(t_i) = ME(t_0) - ME(t_i)$$
(4)

In these expressions, m is the mass of the monitored block (kg);  $v(t_i)$  is the block's velocity at time  $t_i$  (m/s);h(t<sub>i</sub>) is the block's elevation at time  $t_i$  (m); ME(t<sub>0</sub>) is the mechanical energy of the block in its initial state (J). Since the initial velocity is 0, ME(t<sub>0</sub>) reduces to the initial potential energy. Because this study focuses exclusively on energy transfer during the landslide, the potential-energy calculation uses the model's initial height as the reference datum rather than true sea level; only the change in potential energy due to relative displacement is considered.

As shown in Fig6., ten monitoring points were installed in the lower, lower-middle, upper-middle, and upper portions of the slide mass to capture representative kinematics and deposition characteristics of the landslide blocks. In addition, the maximum unbalanced force was tracked to verify that the model reached post-failure stability. The properties of each monitoring point are listed in Table 2.

For the global analysis, the overall displacement, velocity, and stress variations of the entire slide mass were recorded and interpreted in conjunction with the stages of the failure process. For the local analysis, displacement, velocity, and stress histories were extracted for each of the ten monitored blocks in all relevant directions and used, together with the governing equations, to compute the evolution of their energies. Following these energy-evolution calculations and kinematic observations, the sequence of failure and run-out at Mogangling was inferred by integrating the numerical results with field-investigation findings.

Table 2: The basic information of blocks where the 10 monitoring points are located

| Position          | ID          | Coordinate                                           | Volume (m³)                | Monitoring contents                                |  |
|-------------------|-------------|------------------------------------------------------|----------------------------|----------------------------------------------------|--|
| bottom            | 1 2         | (-79,-284,340)<br>(-68,66,355)                       | 939.9<br>1021.6            | Displacement-X                                     |  |
| lower middle part | 3<br>4<br>5 | (-211,-476,474)<br>(-223,-97,465)<br>(-203,224,498)  | 756.2<br>966.8<br>1553.4   | Displacement-Y Displacement-Z Velocity-X           |  |
| upper middle part | 6<br>7<br>8 | (-420,-557,660)<br>(-500,-116,661)<br>(-526,286,692) | 1692.3<br>1754.3<br>1003.5 | Velocity-Y<br>Velocity-Z<br>Stress-XX<br>Stress-YY |  |
| top               | 9<br>10     | (-680,-283,752)<br>(-719,120,755)                    | 856.9<br>967.8             | Stress-ZZ                                          |  |
|                   | 11          |                                                      |                            | Maximum unbalanced force                           |  |

Fig6. Location of 10 monitoring points

# 2.4 Structural extraction of landslide accumulation area

To examine the relationship between the dispersal degree of the landslide deposition zone and the evolution of energy during the sliding process, we quantify dispersion by extracting the volume and surface area of the deposition zone. An alpha-shape algorithm is applied to the slide mass to generate an enclosing envelope, from which its volume and area are computed. To facilitate a direct comparison between the changes in volume and area and the corresponding energy evolution, two dimensionless indices are introduced: volume swelling (VS) and area growth (AG):

$$VS = \frac{V_{t} - V_{0}}{V_{0}} \times 100\% \tag{5}$$

$$AG = \frac{S_t - S_0}{S_0} \times 100\% \tag{6}$$

In these expressions,  $V_0$  denotes the initial volume of the landslide mass (m³);  $V_t$  denotes the volume of the enclosing envelope at time t (m³);  $S_0$  denotes the initial surface area of the landslide mass (m²);  $S_t$  denotes the surface area of the enclosing envelope at time t (m²).

Considering that the choice of the alpha parameter ( $\alpha$ ) in the Alpha Shape algorithm directly affects the accuracy of the envelope fitting as well as the computed volume and surface area, this study uniformly sets the  $\alpha$  value to 1. This approach ensures a tight fit of the envelope model while minimizing geometric deviations introduced by parameter tuning.

Building on this foundation, to further explore the relationship between rock mass fragmentation and energy transfer during the landslide process, the evolution of the landslide system's kinetic energy (KE), potential energy (PE), and dissipated energy (DE) concerning the dimensionless indices volume








swelling (VS) and area growth (AG) is visualized using line plots. Specifically, the horizontal axis represents either VS or AG, while the vertical axis shows the instantaneous values of the three energy components (KE, PE, DE). Each energy curve is distinguished by a unique marker and line style to ensure clarity and facilitate visual comparison. By connecting discrete data points at successive time steps, a continuous evolution trajectory is formed, allowing dynamic trends in energy conversion and dissipation to be observed across different levels of mass dispersion. This visualization approach not only intuitively illustrates the influence of rock fragmentation and dispersal on energy distribution but also provides a quantitative basis for understanding energy transfer mechanisms in the dynamics of landslides.

#### 3 Results

#### 3.1 Motion and deposition of the Mogangling landslide

The results of the numerical simulation successfully reproduced the entire process of the Mogangling landslide. By tracking the cumulative runtime, it was found that the landslide lasted for a total of 131 seconds from initiation to completion. Fig7 illustrates the cloud maps of velocity distribution for the overall model and stress distribution along the sliding plane, revealing the kinematic evolution of the landslide.

Fig7. (a) and (h) show the initial state of the landslide, during which the slope has not yet been affected by seismic forces and the landslide mass has not begun to move.

As shown in Fig7. (b) and (i), during the vibration-induced tension-cracking stage of the landslide, the maximum velocity reached 5.9 m/s. The seismic waves directly triggered vibration and tension failure in the slope rock mass. The first-arriving P-waves caused intense vertical dislocation along the steeply dipping structural planes outside the slope. Subsequently, under the combined action of the P-waves and the later-arriving S-waves, tensile cracking developed along these structural planes. Under the combined influence of seismic wave amplification at higher elevations, the back-slope effect, and dynamic interface stress effects, strong unloading-induced tension cracking occurred in the source area rock mass along pre-existing potential sliding surfaces.

As shown in Fig7. (c) and (j), during the high-speed initiation stage of the landslide, the maximum velocity reached 19.6 m/s. The intense seismic loading led to the formation of tensile cracks along the rear and lateral margins of the landslide. The back-slope effect and amplification at higher elevations during seismic wave propagation significantly increased the peak ground acceleration of the Mogangling slope, triggering the initial movement of the sliding mass. During this initiation process, the dilative expansion along the sliding surface intensified the fragmentation and brecciation of materials along the potential failure planes. This continued until a sudden fracture occurred at the front of the slope, causing the landslide mass to accelerate rapidly.

As shown in Fig7. (d) and (k), during the collapse stage, the maximum velocity reached 27.1 m/s. Under the combined influence of structural plane control and strong seismic forces, the landslide mass was ejected and transformed into a fragmented, granular state. The front edge of the slide subsequently





collided with the bedrock surface of the lower slope. This violent impact caused the already disaggregated material to rapidly disperse, generating fine debris such as rock fragments. As the middle and rear portions of the landslide continued to collapse downslope, ongoing impacts ultimately led to the formation of a loosely consolidated fractured layer approximately 20 meters thick near the interface between the landslide deposit and the underlying bedrock.

As shown in Fig7. (e) and (l), during the impact-at-base stage, the maximum velocity reached 31.6 m/s.

After a brief airborne trajectory, the entire landslide mass collided with the ground. The immense impact energy caused both the sliding mass and the adjacent bedrock near the basal interface to disintegrate and collapse, producing fragmented rock debris. Simultaneously, the landslide mass instantly broke apart and disintegrated upon impact, with only a small portion retaining original rock characteristics, forming a layered pseudo-bedrock. The disintegrated landslide material then transformed into a debris flow, which, driven by substantial downslope inertia, continued to move rapidly toward the Dadu River valley. The velocity reached its peak upon arrival at the valley floor.

As shown in Fig7. (f) and (m), during the impact and disintegration stage, the velocity began to decrease to 25.7 m/s. Upon reaching the Dadu River valley, the rock mass collided with the valley bottom. The velocity of the debris flow front suddenly dropped, while the debris flow started to spread laterally towards both sides. Additionally, due to the relatively flat and open terrain of the downstream valley, a pronounced "protrusion" facing upstream appears on the Huashibao terrace across the river. Consequently, under the combined effects of topographic obstruction and guidance, the debris flow's movement shifted slightly towards the downstream side, eventually forming a dam-like accumulation slightly offset downstream. The existing deposits still reflect this flow pattern.

As shown in Fig7. (g) and (n), during the river-blocking damming stage, the velocity gradually approached 0 m/s. The rapidly collapsing landslide mass disintegrated into a debris flow that surged downstream at high speed, instantly blocking the Dadu River and forming a natural landslide dam. Due to limitations of the discrete element numerical simulation, the subsequent dam-break process after river blockage was not modeled. However, based on the simulation results, it can be inferred that the landslide-induced damming caused a rapid rise in the water level behind the dam, resulting in backflow flooding in the surrounding counties and cities.

Fig7. (o) presents compelling evidence of agreement between the numerical simulation results and the actual field observations. Although the landslide in this region occurred in 1786, and significant topographic alterations have resulted from centuries of weathering, several clear points of correspondence can still be identified. For instance, a protruding ridge divides the source area into two distinct sections—a feature that is also reproduced in the numerical simulation. Furthermore, the extent and morphology of the accumulated landslide material in the simulation closely match those observed in reality. These consistencies support the validity and practical relevance of the numerical simulation.

Fig7. Simulation results of Mogangling landslide: (a-g) cloud map of velocity when t = 0s, 3s, 7s, 9s, 12s, 15s, 131s; (h-n) cloud map of stress when t = 0s, 3s, 7s, 9s, 12s, 15s, 131s; (o) Validation of numerical simulation results against real-world evidence.

## 3.2 Energy evolution of the landslide





#### 3.2.1 Energy evolution for the whole debris mass

During the numerical simulation, block parameters of the entire landslide mass, including volume, velocity, and displacement, were extracted to calculate the energy evolution throughout the landslide process. Fig8. illustrates the energy variations of the entire sliding mass during the event. It is evident that, due to the law of energy conservation, the total sum of the three forms of energy remains constant throughout the landslide.

Initially, the rock mass had not started to slide and therefore stored only potential energy. Between 0 and 3 seconds, slight deformation occurred without significant damage; during this period, only a small amount of potential energy was converted into kinetic energy, with some energy dissipated due to friction. From 3 to 12 seconds, the landslide experienced rapid initiation, and the entire sliding mass abruptly descended along the sliding plane. The kinetic energy increased steadily, reaching its peak. Around 13 seconds, the sliding mass impacted the riverbed, causing a sharp drop in kinetic energy due to the obstruction of rock blocks.

Subsequently, the upper portion of the landslide continued to move downward until coming to a complete stop. Throughout the process, changes in the overall kinetic energy of the landslide correspond well with the velocity variations of the rock mass shown in Fig7. Dissipated energy continuously increased while potential energy decreased, consistent with physical principles. This can be regarded as a successful energy extraction numerical simulation.



Fig8. Evolution of energy during the landslide

# 3.2.2 Energy evolution for individual rock blocks

The dynamic parameters of 10 monitoring points were recorded to determine the energy of individual blocks at different locations. As shown in Fig9., the energy variation curves indicate that, unlike the total landslide mass, the energy of single blocks is not conserved. This is because each block is subjected to compression and pushing forces from surrounding blocks, which alter its mechanical energy.

Notably, some blocks—such as Block 1 and Block 3—exhibit a rebound in potential energy after it approaches zero. This can be explained by these blocks bouncing to the opposite bank after impacting the base, increasing their potential energy. Instances where dissipated energy appears to decrease are actually due to the calculation method used. According to the calculation approach, it can be inferred that during these periods, the monitored blocks received collisions from neighboring blocks that caused a temporary increase in the combined sum of potential and kinetic energy.

To further analyze these phenomena, we combined the 3D trajectory plots of the 10 blocks with the identification of effective collisions.

Fig9. Energy analysis for 10 individual blocks in Mogangling landslide: (a) energy evolution of point 1; (b) energy evolution of point 2; (c) energy evolution of point 3; (d) energy evolution of point 4; (e) energy evolution of point 5; (f) energy evolution of point 6; (g) energy evolution of point 7; (h) energy evolution of point 8; (i) energy evolution of point 9 and (j) energy evolution of point 10.

- Fig10. (a) shows the trajectories of the 10 monitored blocks, while Fig10. (b) presents the frequency of effective collisions experienced by blocks at different locations during the landslide process. It can be observed that Blocks 1 and 3 exhibit bouncing rebounds, confirming the previous inference. Each monitored block at different locations displays distinct movement characteristics.
  - Combining this with Fig10. (b), it is clear that as the migration distance increases during the landslide, the number of collisions also rises continuously, reaching a peak once the blocks enter the river channel.
- At this stage, the frequency at which individual blocks receive energy transfer from surrounding blocks also increases.

Fig10. (a) movement trails of 10 monitoring points and (b) the plot of travel distance vs. collision frequency.

# 425 3.4 Structural evolution as a function of energy change

This study utilized numerical simulation data of the landslide to perform Alpha Shape envelope fitting on the deposit mass at 10 representative time points, extracting the corresponding envelope volumes and surface areas (Fig11.). It can be observed that as the simulation progresses, both the volume and surface area of the deposit region exhibit a continuous increasing trend. This effectively reveals the coupled evolution characteristics between debris production and dispersion degree, laying a solid foundation for subsequent quantitative analysis of energy transfer and dissipation mechanisms.

Fig11. Comparison of the overall envelope fitting of the landslide at 10 different times and the shape of the landslide mass in numerical simulation: (a-i) when time = 0s, 3s, 7s, 9s, 12s, 15s,32s,71s, 100s, 131s.

- For the fitted envelope, we first extracted its volume (V) and surface area (A) at each time point, based on which the volume expansion rate (VS) and surface area growth rate (AG) were calculated. The landslide's kinetic energy (KE), potential energy (PE), and dissipated energy (DE) were then plotted as functions of VS and AG (Fig12.). It is observed that when the variation amplitude of KE increases, the rate of increase in VS accelerates, and the growth magnitude of AG also significantly rises.
- This phenomenon can be interpreted as follows: during the landslide development, blocks continuously undergo compression and collisions, transferring kinetic energy from surrounding blocks to the target block. In the process of energy transfer, collisions and friction cause block fragmentation and disintegration, producing large amounts of debris. These newly generated fragments fill the interior and surface of the deposit mass, causing a sudden expansion of the envelope volume and a sharp increase in surface area.
  - Simultaneously, the sharp rise of DE shown in the figure further confirms significant fragmentation and disintegration in the monitored block group, with a rapid increase in the degree of fragmentation. When

the volume expansion rate reaches approximately 29.47% and the surface area growth reaches about 319.5%, it can be considered that the landslide transitions from a solid phase to a fluidized phase. At this point, the kinetic energy peaks, indicating an increased collision frequency within the landslide, which correspondingly induces more debris production.

Fig12. Overall energy variation of the sliding mass: (a) as a function of volume; (b) as a function of area.

### 460 4 Conclusions

Based on the analysis of the pre-landslide topography restoration and the numerical simulation results presented above, the following conclusions can be drawn:

- 1. Based on the continuity of topographic contours, a preliminary restoration of the ancient landslide can be performed. It should be noted that:(a) Outside the landslide boundaries, there exist contour lines consistent with the continuity of the mountain range, and their trajectory equations can be extracted as a basis for restoring the contours;(b) The landslide deposit area is often difficult to restore due to the lack of surrounding reference features. In such cases, restoration can be guided by referencing other mountain areas within the same stratigraphic unit.
- 2. Discrete element numerical simulation can incorporate terrain, stratigraphy, geological structures, and boundary conditions, allowing for the extraction of kinetic energy, potential energy, and dissipated energy of monitored blocks. Comparison with field imagery and result analysis ensures that the simulation outcomes align with actual conditions. Based on numerical simulation results and field investigations, the landslide movement process can be analyzed. Furthermore, effective block collisions during the landslide can be identified by extracting local peaks from displacement-velocity curves.
  - 3. Based on the energy analysis of both the entire landslide mass and individual blocks, the following conclusions can be drawn: for the landslide as a whole, the conversion among the three types of energy obeys the law of energy conservation—that is, dissipated energy continuously increases, and the sum of the three energies equals the initial potential energy before sliding; for individual landslide blocks,

490

495

- energy is not conserved due to collisions and compressions from surrounding blocks transferring energy, which is reflected by temporary decreases in the dissipated energy of certain blocks at specific times.
  - 4. Based on the analysis of effective collisions within the entire landslide mass, it can be concluded that during the collapse stage, the front portion of the rock mass experiences more collisions than the rear portion, with a clear trend of decreasing collision frequency toward the rear. When the rear rock mass collides with the front, its dissipated energy sharply increases and transfers energy to the front rock mass, enabling the front portion to slide farther.
  - 5. By comparing the volume expansion and surface area growth of the landslide envelope with the energy variations, it can be concluded that during energy transfer, collisions and friction cause blocks to fracture and disintegrate, generating a large amount of debris. These newly formed fragments fill the interior and surface of the deposit mass, resulting in a sudden expansion of the overall envelope volume and a sharp increase in surface area.

#### **Statements and Declarations**

Code availability. The codes that support the findings of this study are available from the first author, Yunfeng Ge, upon reasonable request.

Authors' contributions. YG: Funding acquisition, Conceptualization, Data curation, Formal analysis, Investigation, Methodology, Resources, Supervision, Validation, Visualization, Writing-original draft; BH: Data curation, Software, Investigation, Validation, Visualization, Writing-original draft; HT: Project administration, Supervision; XF: Data curation, Investigation. LZ: Data curation, Investigation.

Competing interests. The authors declare that they have no conflict of interest.

Acknowledgments/funding. This work was supported by the National Natural Science Foundation of China (No. 42477177), the Science and Technology Planning Project of Kunyu City in 14th Division (No. 2025-14-01), and the Hubei Natural Science Foundation Joint Fund Project (No. 2024AFD005). We thank editors and anonymous reviewers for their valuable comments.

Availability of data and material. The data and materials that support the findings of this study are available from the first author, Yunfeng Ge, upon reasonable request.

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
