# Peer review of "Energy and structural evolution process of high-altitude and long-runout landslides induced by a strong earthquake"

_EGUsphere, 2025_

## Author Comment (AC1)

Responses to Reviewer #1

**Specific comments:**

1. Dear editor, dear authors,

first, there is a structural problem as you include side description and model setup in the methodology section.

To solve this problem alone .. a medium to major revision would have been necessary - this requires a resubmission.

**Response**: We sincerely thank the reviewer for pointing out this structural flaw. We fully agree that the organization of the manuscript needed improvement to meet high academic standards. Action taken: In the revised manuscript, we have restructured the article significantly.

1. We have extracted the geological background and site description from the original Methodology section and created a new independent section titled "**2 Study Area and Geological Background**".

2. The "**3 Methodology**" section now strictly focuses on the technical approaches, including the contour restoration method, the 3DEC numerical simulation principles, and the energy calculation equations.

3. The numerical model setup (mesh generation, boundary conditions) is presented in a dedicated subsection within the Methodology, clearly separated from the geological description.

2. However, when I see you back-analysis and pre-failure slope reconstruction, I have the impression that you just worked on the scarp part and not on the part where there are still millions of cubic meters of landslide material that you just left in your pre-failure model.

This is not correct, you have to remove this material as it constitutes part of the failure zone. (noting that for sure a large part of the material has been removed by Dadu River,but as I wrote above, at least 10Mio m$^3$ are still on-site ... I was there in 2023).

For this full reconstruction in 3D please check paper Mreyen et al. 2022 (doi: 10.1016/j.enggeo.2022.106774).

When you resubmit your manuscript, also show 2D sections comparing each pre-failure model with the post-failure model section.

yours

reviewer H

**Response**: We appreciate the reviewer's keen observation and their on-site experience at the Mogangling landslide. We understand the concern that failing to remove the existing landslide deposit (accumulation area) would lead to an incorrect sliding bed geometry.

We would like to clarify that removing the landslide deposit was indeed a core step in our reconstruction process, although the textual description and figures in the original manuscript might not have highlighted this sufficiently.

1. Clarification on Deposit Removal: As indicated in our Flow Chart (Fig. 1) under the "Lower Slide Bed Restoration" step, we explicitly performed "Deposit Removal". Our method involved identifying the boundary of the accumulation area and modifying the contour lines. By straightening the convex contour lines caused by the accumulation and aligning them with the stable bedrock contours on both flanks, we logically "excavated" the deposit volume to restore the pre-failure valley topography. We would like to direct your attention to Fig. 3(c) in the manuscript, which illustrates the comparison of the surface topography before and after restoration.

[Figure]

Fig3. (c) Comparison map of surface topography before and after restoration

[Figure]

2. New 2D Comparison Sections: Following the reviewer's specific request, we have added a new figure (Fig.3 d in the revised manuscript) displaying multiple 2D cross-sections. These sections clearly overlay the post-failure topography (current surface) and our reconstructed pre-failure topography (sliding bed). The area between these two lines clearly demonstrates the volume of the material (the deposit) that was removed in our model to reconstruct the original slope.

[Figure]

Fig. 3 (d) Terrain restoration comparison map

3. Regarding the basis for the terrain restoration: We have carefully read the recommended paper (Mreyen et al., 2022). It provides an excellent framework for 3D reconstruction. We have cited this work in our revised methodology section to better contexturalize our contour restoration approach within the current state of the art.

---

## Author Comment (AC2)

Responses to Reviewer #2

**General Comments:**

This manuscript provides a systematic investigation of the dynamics of the strong earthquake-induced Mogangling paleo-landslide by integrating topographic reconstruction, discrete element simulation, energy analysis, and Alpha Shape-based morphological quantification. The research demonstrates methodological innovation and in-depth analysis. Its outstanding contribution lies in proposing a quantitative method for identifying "effective collisions" and successfully revealing the intrinsic relationship between energy evolution and the structural fragmentation of the landslide mass (i.e., volume expansion and surface area growth). This offers a novel perspective for understanding the energy and structural evolution processes of high-altitude, long-runout landslides. Although certain limitations exist, the robustness of the conclusions has been ensured through sensitivity analysis and other means. The study holds significant theoretical value and practical implications for risk assessment of such landslides, representing a comprehensive and solid piece of outstanding work.

In summary, I see potential in the manuscript and it may eventually meet standards of Natural Hazards and Earth System Sciences after a moderate revision after addressing my concerns listed below.

**Response:** We would like to express our sincere gratitude for your encouraging evaluation and the time you took to review our manuscript. We are particularly heartened by your recognition of the methodological innovations in this study, especially regarding the quantitative identification of "effective collisions" and the analysis of the intrinsic relationship between energy evolution and structural fragmentation. Your positive feedback has given us great confidence in the value of this work.

We have carefully studied the specific concerns listed below and have made corresponding revisions to further improve the quality and rigor of the manuscript.

Please find our point-by-point responses to your comments below.

**Specific comments:**

1. Some figure titles could be shortened to enhance clarity.

**Response:** We agree with the reviewer that concise figure titles significantly enhance clarity and readability. We have carefully reviewed all figure captions and simplified those that were overly wordy or repetitive without losing necessary information.

| Figure | Original Caption | Revised Caption |
|--------|------------------|-----------------|
| Fig. 5 | Effective collisions identification: (a) definition of effective collisions; (b) threshold=10%; (c) threshold=20%; (d) threshold=30%; (e) threshold=40% and (f) threshold=50%. | Effective collisions identification: (a) definition of effective collisions; (b-f) threshold=10%, 20%, 30%, 40% , 50%. |
| Fig. 9 | Energy analysis for 10 individual blocks in Mogangling landslide: (a) energy evolution of point 1 ; (b) energy evolution of point 2 ; (c) energy evolution of point 3 ; (d) energy evolution of point 4 ; (e) energy evolution of point 5 ; (f) energy evolution of point 6 ; (g) energy evolution of point 7 ; (h) energy evolution of point 8 ; (i) energy evolution of point 9 and (j) energy evolution of point 10. | Energy analysis for 10 individual blocks in Mogangling landslide: (a-j) energy evolution of point 1, 2, 3, 4, 5, 6, 7, 8, 9, 10. |

2. The physical and mechanical parameters assigned to the Discrete Element Method (DEM) model should be clearly documented with their sources.

**Response:** We thank the reviewer for emphasizing the importance of parameter

documentation. In the revised manuscript, we have clarified the sources for the physical and mechanical parameters in Section 2.1 and the caption of Table 1. Specifically, we have detailed that:

1. Basic physical parameters were obtained from laboratory tests on rock samples collected from the site.

2. Rock mass strength parameters were estimated using the Hoek-Brown criterion based on field geological survey data and empirical values for similar granite lithologies in the Dadu River region.

Citations for the empirical values and comparative studies have been added to the revised text and table caption to ensure reproducibility.

**References:**

Wu, H., Shi, A., Ni, W., Zhao, L., Cheng, Z., and Zhong, Q.: Numerical simulation on potential landslide–induced wave hazards by a novel hybrid method, Eng. Geol., 331, 107429, https://doi.org/10.1016/j.enggeo.2024.107429, 2024.

Wu, J., Wang, Y., Dong, S., Chen, Y., and Wang, L.: Genetic Mechanism and Failure Process of the Mogangling Seismic Landslide, J. Geol. Soc. India, 82, 277-282, https://doi.org/10.1007/s12594-013-0150-3, 2013.

Zhao, B., Wang, Y., Wu, J., Su, L., Liu, J., and Jin, G.: The Mogangling giant landslide triggered by the 1786 Moxi M 7.75 earthquake, China, Nat. Hazards, 106, 459-485, https://doi.org/10.1007/s11069-020-04471-1, 2021.

3. The rationale behind the 30% threshold for defining effective collisions requires further elaboration.

**Response:**We appreciate the reviewer's request for further clarification on this critical parameter. The selection of the 30% threshold was not arbitrary but was determined through a systematic sensitivity analysis aimed at optimizing the Signal-to-Noise

Ratio (SNR) in the velocity data. In Discrete Element Method (DEM) simulations, block velocities often exhibit minor high-frequency fluctuations due to numerical contact adjustments and friction, which do not represent physical collisions. As shown in Fig. 5, the lower thresholds (10%–20%) were too sensitive, capturing these minor numerical fluctuations (noise) as false positives, leading to an overestimation of collision frequency. These higher thresholds were overly stringent, filtering out legitimate impact events that involved significant energy transfer but lower instantaneous acceleration peaks (false negatives).

Therefore, the 30% threshold was identified as the optimal balance point where numerical noise is effectively filtered out while significant kinematic changes induced by physical impacts are accurately retained. We have expanded the explanation in Section 2.2 of the revised manuscript to articulate this rationale more clearly.

[Figure]

[Figure]

Fig5. Effective collisions identification: (a) definition of effective collisions; (b-f) threshold=10%, 20%, 30%, 40%, 50%.

4. The conclusion should be started with a short paragraph briefly explaining the study's topic, contribution, and methodology. Then, present the main findings as clear and concise bullet points.

**Response:** We appreciate the reviewer's constructive suggestion regarding the structure of the conclusion. We have rewritten the Conclusion section to strictly follow the recommended format. In the revised manuscript, we have added an introductory paragraph that summarizes the study's topic (the Mogangling landslide), its main contributions (energy-structure coupling analysis), and the methodology used (topographic reconstruction and DEM simulation). Following this summary, the specific findings are presented as clear and concise bullet points:

**4. Conclusions**

This study investigated the dynamic evolution and energy transfer mechanisms of the Mogangling high-altitude long-runout landslide triggered by the 1786 Kangding earthquake. By integrating pre-landslide topographic reconstruction based on contour continuity, 3D discrete element numerical simulation and Alpha Shape structural quantification, we successfully reproduced the landslide process. This work establishes a quantitative framework linking microscopic block collisions to macroscopic morphological changes, providing a theoretical basis for analyzing

energy conversion in complex rock avalanches.

5. The typesetting of equations in the manuscript could be further standardized. For instance, variables should be italicized, while mathematical operators should remain upright. As shown in Equation (2), the mass m and velocity v should be italicized.

**Response:** We thank the reviewer for their meticulous attention to typesetting details. We have thoroughly reviewed and standardized all mathematical equations and inline math symbols throughout the manuscript.

6. The manuscript contains a considerable number of physical variables in its equations. To facilitate reader reference and ensure clarity and consistency in terminology, it is recommended to add a Nomenclature section before the main text or in an appendix. This table should list all variable symbols, their corresponding physical meanings, and units, which will significantly enhance the readability and standardization of the paper.

**Response:** We fully agree with the reviewer's recommendation. Given the number of physical variables involved in the energy and structural evolution analysis, a dedicated nomenclature section is essential for clarity and consistency. We have added a Nomenclature section (placed as Appendix A in the revised manuscript) that lists all variable symbols, their physical meanings, and corresponding units. This addition ensures standard terminology and enhances the overall readability of the paper.

Appendix A: Nomenclature

| Symbol | Physical Meaning | Unit |
|--------|------------------|------|
| **Energy Parameters** | | |
| ME | Mechanical Energy | J |
| KE | Kinetic Energy | J |
| PE | Potential Energy | J |

| | | |
|---|---|---|
| DE | Dissipated Energy | J |
| $t_i$ | Time instance i | s |
| $t_0$ | Initial time (t=0) | s |
| **Block Kinematics** | | |
| m | Mass of the monitored block | kg |
| v | Velocity of the block | m/s |
| h | Elevation (height) of the block | m |
| g | Gravitational acceleration | $m/s^2$ |
| **Structural Evolution** | | |
| VS | Volume Swelling rate | % |
| AG | Area Growth rate | % |
| $V_t$ | Volume of the landslide envelope at time t | $m^3$ |
| $V_0$ | Initial volume of the landslide mass | $m^3$ |
| $S_t$ | Surface area of the landslide envelope at time t | $m^2$ |
| $S_0$ | Initial surface area of the landslide mass | $m^2$ |
| α | Alpha parameter for Alpha Shape algorithm | - |

7. The selection of the Luding station record from the Wenchuan earthquake as input for the 1786 Kangding earthquake represents a reasonable alternative. However, could you provide a more detailed explanation regarding the similarities in source mechanism, epicentral distance, and propagation path between the two events to further strengthen the justification for this substitution?

**Response:** We thank the reviewer for this insightful suggestion. In the revised Section

3.1, we have added a detailed explanation covering the source mechanism, propagation path, and site effects. The justification is based on three key similarities:

1. Tectonic Affinity: Both earthquakes occurred within the Y-shaped fault zone system at the eastern margin of the Tibetan Plateau, sharing similar thrust-strike-slip stress regimes and crustal rupture characteristics.

2. Site Response Consistency: The Mogangling landslide and the Luding recording station are both located in the Dadu River valley . Using the record from Luding station captures the specific valley-site effects and topographic amplification characteristics unique to this high-relief canyon terrain, which are critical for slope stability analysis.

[Figure]

Fig. 11 The location of Luding County and Mogangling landslide (Zhou et al., 2024)

Reference:

Zhou, H., Ye, F., Fu, W., Liu, B., Fang, T., & Li, R. (2024). Dynamic effect of landslides triggered by earthquake: A case study in Moxi Town of Luding County, China. Journal of Earth Science.

8. The description of the regional geotechnical conditions in Section 2.1 "Geo-mechanical model building" could be condensed without compromising academic rigor. It is recommended to streamline the content by focusing on the key geological features directly relevant to the model construction, while retaining all critical data and parameters necessary for reproducibility.

**Response:** We appreciate the reviewer's suggestion to improve the conciseness of the manuscript. To accommodate a structural suggestion from another reviewer, we have extracted the general background description and placed it into a newly dedicated section: "**2 Study Area and Geological Background**" Consequently, the revised Section 2.1 has been significantly condensed. It now strictly focuses on the geological features and parameters explicitly implemented in the 3DEC model (e.g., specific lithology and fault geometry), fulfilling your recommendation to focus on key features while retaining critical data for reproducibility.

---

## Author Comment (AC3)

Responses to Reviewer #3

**General Comments:**

Here is the major revision advice in English, presented as a single continuous paragraph (no paragraph breaks), matching your requirements: The manuscript presents valuable work, but it requires substantial restructuring to improve logical clarity and academic rigor. A fully independent and substantially enriched Discussion section must be added, as many interpretative statements are currently embedded within the Results and should be relocated and expanded. This new Discussion should systematically address:

**Response:** We explicitly thank the reviewer for the comprehensive evaluation and the detailed roadmap for restructuring the manuscript. We fully agree that the previous structure limited the presentation of our findings and that a substantial reorganization was necessary to improve logical clarity and academic rigor. In accordance with your advice, we have performed a major structural overhaul of the manuscript: 1. Discussion Section: We have established a new, fully independent "**5 Discussion**". We removed interpretative statements from the Results section and expanded them into a deep analytical discussion. As requested, this new section is systematically divided into four subsections addressing: "**5.1 Dynamic evolution and energy transfer mechanism**"; "**5.2 Structural fragmentation and solid-to-fluid phase transition**"; "**5.3 Uncertainties and limitations of the numerical modeling**"; and "**5.4 Implications for hazard-chain evolution**"

**Specific comments:**

1. comparison with existing studies on high-altitude long-runout landslides, energy transfer, fragmentation, and fluidization mechanisms;

**Response:** We appreciate the reviewer's suggestion. We have systematically addressed these comparisons in the newly established "**5 Discussion**": 1. In **5.1 Dynamic evolution and energy transfer mechanism**: We compared the kinematic

behavior of the Mogangling landslide with the well-documented Donghekou landslide triggered by the 2008 Wenchuan earthquake (e.g., Sun et al., 2011). The discussion confirms that both events share the typical "high-speed ejection and collision-disintegration" pattern characteristic of high-altitude landslides under strong seismic loading. And we contextualized our finding of "rear blocks transferring energy to frontal blocks" within the classic momentum transfer theory of rock avalanches (e.g., Heim, 1932; Davies, 1982). We argued that the non-conservation of energy at the local block scale—driven by effective collisions—provides a physical explanation for the "pushing effect" that enables the frontal mass to achieve excessive runout distances; 2. In **5.2 Structural fragmentation and solid-to-fluid phase transition**: We contrasted our method with traditional qualitative descriptions of disintegration. By citing granular flow theories ( Iverson, 1997), we highlighted the innovation of using Alpha Shape-derived indices (VS and AG). We proposed that the identified thresholds (VS > 29.47%) serve as a quantitative metric for the solid-to-fluid phase transition, offering a more precise tool for analyzing the fluidization process than previously available.

2. key uncertainties, including the smoothing effect of the contour-restoration method, the limitations of using Wenchuan earthquake records as a proxy for the 1786 Kangding event, the influence of uniform block size in 3DEC, the simplified treatment of structural planes, and numerical constraints in representing debris-flow–like behavior;

**Response:** We sincerely thank the reviewer for summarizing these critical uncertainties.

We have addressed these points systematically in the new **5.3 Uncertainties and limitations of the numerical modeling**:

**1. the smoothing effect of the contour-restoration method:**

We acknowledged that the restoration method based on contour continuity inevitably smooths out micro-topography. However, we argued that for a landslide of this magnitude, the global runout path and energy evolution are primarily controlled by

the valley-scale topography rather than local surface roughness;

**2. the limitations of using Wenchuan earthquake records as a proxy for the 1786 Kangding event:**

We justified the use of the 2008 Wenchuan earthquake record (Luding station) based on two key similarities: (a) Tectonic Affinity: Both events occurred within the same Xianshuihe-Longmenshan fault system; (b) Site Effects: The Luding station is located in the Dadu River canyon, similar to the Mogangling site. Using this record inherently preserves the specific valley-site effects (topographic amplification) that synthetic waves might miss;

[Figure]

Fig. 11 The location of Luding County and Mogangling landslide (Zhou et al., 2024)

**3. the influence of uniform block size in 3DEC:**

We clarified that while a uniform block size distribution simplifies the internal interaction, it was a necessary compromise for computational efficiency. We

emphasized that this setup is sufficient to capture the macroscopic "effective collision" trends and the momentum transfer mechanism, which are the main focus of this study.

**4. the simplified treatment of structural planes:**

We clarified that the simplification was not arbitrary but data-driven. By employing automatic discontinuity identification technology on the field survey data, we statistically identified three dominant joint sets (as shown in the figure below) that control the rock mass stability. The 3DEC model explicitly incorporates these three critical sets. While minor random fissures were simplified to optimize computational efficiency, this approach accurately captures the primary failure mechanism—sliding along bedding planes and cutting through the major joints—without compromising the macroscopic kinematic behavior;

[Figure]

Fig4. Discrete numerical model with fine topography: (b) Structural surface model

**5. numerical constraints in representing debris-flow–like behavior:**

We clarified that the 3DEC model simulates dry granular flow rather than

water-saturated debris flow. We acknowledged that the model does not account for pore water pressure or fluid coupling. However, we argued that the observed high mobility is driven by mechanical fluidization—a state where high-frequency collisions between fragmented blocks generate dispersive stresses, reducing bulk friction. This approach is widely accepted in rock mechanics for simulating the kinematic behavior of rock avalanches before they fully enter water bodies.

Reference:

Zhou, H., Ye, F., Fu, W., Liu, B., Fang, T., & Li, R. (2024). Dynamic effect of landslides triggered by earthquake: A case study in Moxi Town of Luding County, China. Journal of Earth Science.

3. deeper interpretation of physical mechanisms such as effective collision frequency, transitions from solid to granular–fluidized motion, the significance of the observed VS and AG thresholds, and the implications for landslide dynamics under strong seismic loading;

**Response:** We appreciate the reviewer's guidance to deepen the physical interpretation of our data. We have significantly enriched the interpretative content in **5.1 Dynamic evolution and energy transfer mechanism** and **5.2 Structural fragmentation and solid-to-fluid phase transition:**

1. **Effective collision frequency**: The role of effective collision frequency in energy redistribution: Our analysis reveals a fundamental link between the frequency of effective collisions and the dynamics of energy transfer. The collision frequency should be interpreted not merely as a kinematic statistic, but as a quantitative proxy for the rate of energy exchange between rock blocks. As shown in Fig. 5, the spatial distribution of collision frequency is highly heterogeneous. The rear and middle sections of the landslide mass exhibit significantly higher collision frequencies compared to the frontal margin. From the perspective of energy evolution, frequent collisions act as high-efficiency conduits for kinetic energy transfer. During the

high-speed propagation stage, the rear blocks, possessing high gravitational potential energy, continuously impact the blocks ahead. This process creates a clear energy transfer pathway: through high-frequency effective collisions, the rear blocks act as energy donors, transferring their momentum and kinetic energy to the frontal blocks . This mechanism explains the observed energy evolution curves , where the kinetic energy of rear blocks fluctuates and dissipates rapidly after impacts, while the frontal blocks maintain high velocities. Consequently, the high collision frequency in the main body serves as the internal engine that sustains the hyper-mobility of the landslide front, driving the excess runout distance characteristic of the Mogangling event.

[Figure]

[Figure]

(e)           (f)

Fig5. Effective collisions identification: (a) definition of effective collisions; (b-f) threshold=10%, 20%, 30%, 40%, 50%.

2. **Fluidized motion**:We deepened the interpretation to explicitly link the phase transition to the high-speed long-runout mechanism. We argued that the transition from a coherent solid block to a granular flow (marked by VS > 29.47%) is the physical cause of bulk friction reduction. This "solid-to-fluid" transformation fundamentally alters the energy dissipation mode, allowing the landslide mass to overcome basal resistance and achieve a runout distance that exceeds standard frictional limits;

3. **The significance of the observed VS and AG thresholds**: We interpreted these thresholds as the critical quantitative boundary that distinguishes the solid-phase sliding regime from the granular-fluidized flow regime. Action taken: To illustrate this quantitative description visually, we added a new schematic figure (Fig. 13 in the revised manuscript). The figure contrasts the coherent solid state below the thresholds (Fig. 13a) with the fragmented granular state above the thresholds (Fig. 13b), explicitly linking the physical morphology with the quantitative VS and AG indices defined in our study;

[Figure]

Fig. 13 Schematic illustration of the solid-to-granular phase transition: (a) the coherent solid phase: Governed primarily by sliding friction exhibiting relatively low sliding velocities; (b) the dispersed granular-fluidized phase: Dominated by rolling friction with a relatively high sliding velocity.

4. **The implications for landslide dynamics under strong seismic loading**:We have rewritten this section to highlight how the study quantitatively characterizes the change in motion state through our two core methodological innovations: Energy Extraction and Structural Analysis.

We explained that strong seismic loading drives the structural disintegration quantified by VS/AG thresholds, which in turn triggers the momentum transfer mechanism quantified by energy evolution. This coupling is what forces the landslide to transition from a solid phase to a fluidized phase. The study shows that the fluidization resulting from this energy-structure interaction significantly amplifies the runout distance. Therefore, accurate hazard forecasting requires considering the degree of seismic-induced fragmentation, as this structural degradation is the primary determinant of the extended disaster scope.

4. broader implications for hazard-chain evolution and future modeling improvements.

**Response:** We thank the reviewer for encouraging us to expand the discussion to broader implications. We have added a new subsection **5.4 Implications for**

**hazard-chain evolution** to the **Discussion**. We linked our simulation results to the catastrophic "**strong earthquake–landslide–impulse waves–damming–outburst flood**" hazard-chain observed in the 1786 event. This study focuses on the process by which seismically-induced landslides disintegrate into debris flows and impact river channels. We emphasized that the dynamic parameters extracted from our model (impact velocity, deposit distribution) are critical inputs for evaluating the safety of current hydropower projects in the Dadu River basin against surge waves and dam-breach floods.

5. The Introduction should be reorganized to emphasize the scientific gap and the methodological innovation of combining contour-continuity restoration, discrete-element energy extraction, and Alpha-shape–based structural evolution analysis. The Methods section should be rewritten with clearer hierarchy and parallel subsections, providing stronger justification for key modeling choices. The Results section should focus strictly on observational outputs, reorganized by landslide stages, while removing mechanistic explanations that belong in Discussion. The Conclusions should be condensed and rewritten after restructuring, highlighting scientific insights rather than descriptive summaries. Language throughout the manuscript requires refinement to improve clarity, remove redundancy, and strengthen scientific precision. Overall, substantial restructuring and expansion of the Discussion—with clear thematic subsections and deeper analytical content—is essential to elevate the manuscript to publication level.

**Response:** We sincerely thank the reviewer for this comprehensive roadmap to elevate the quality of our manuscript. We have rewritten the **Introduction, Methodologies,Results and Discussions** to sharpen the focus on the scientific gap and our innovation.

**1. Introduction**

We have rewritten the Introduction to clearly define the scientific gap: the lack of quantitative links between microscopic energy evolution and macroscopic structural changes in paleo-landslides. We explicitly positioned our "Triad Approach"

(Contour-continuity restoration + Discrete element energy extraction + Alpha-shape structural quantification) as the core methodological innovation filling this gap.

**2. Methodologies**

(1) Section **2.1** is now dedicated solely to "**2. Study Area and Geological Background**"

(2) Section 3 is strictly for "Methodology", with parallel subsections providing strong justifications for key modeling choices (e.g., the rationale for using the Luding seismic record and the parameter calibration process).

**3. Results**

We have rigorously stripped the **Results** (Section 4) of all interpretative and mechanistic explanations. The section is now organized strictly by landslide stages, reporting only observational outputs (velocity fields, energy curves, and structural indices). All mechanistic interpretations have been relocated to the **Discussion**.

**4. Discussions**

As detailed in our responses to previous comments, we have established a new, fully independent **Discussion** (Section 5) with four thematic subsections. This section now systematically addresses comparisons with existing studies, key uncertainties, physical mechanisms (phase transition), and broader hazard implications.

**5. Conclusions**

We have completely rewritten and condensed the Conclusions. We moved away from a descriptive summary of the simulation steps and instead highlighted the core scientific insights derived from the restructured discussion.